# Pet Owners’ Preferences for Quality of Life Improvements and Costs Related to Innovative Therapies in Feline Pain Associated with Osteoarthritis—A Quantitative Survey

**DOI:** 10.3390/ani14162308

**Published:** 2024-08-08

**Authors:** Andrea Wright, Edwina Gildea, Louise Longstaff, Danielle Riley, Nirav Nagda, Kristina DiPietrantonio, Ashley Enstone, Robin Wyn, David Bartram

**Affiliations:** 1Zoetis, 10 Sylvan Way, Parsippany, NJ 07054, USA; 2Zoetis UK Ltd., First Floor, Birchwood Building, Springfield Drive, Leatherhead KT22 7LP, UK; edwina.gildea@zoetis.com (E.G.);; 3Adelphi Values PROVE, Adelphi Mill, Bollington SK10 5JB, UK; 4Outcomes Research, Zoetis International Operations, Loughlinstown, D18 T3Y1 Dublin, Ireland

**Keywords:** osteoarthritis, pain, pet owner preference, quality of life, willingness to pay

## Abstract

**Simple Summary:**

This research examined UK cat owners’ preferences for treatments for feline osteoarthritis, including preferences for quality of life improvements, safety considerations, and costs associated with hypothetical new pain therapies. Aspects of quality of life from previous research were extracted and included in exploratory interviews with cat owners (*n* = 3) to identify the key domains that contribute to the quality of life of cats. Descriptions of the quality of life of cats with osteoarthritis and hypothetical product characteristics were developed and validated through interviews with veterinarians (*n* = 3). An online survey was subsequently shared with 255 pet owners in the UK. Pet owners were presented with quality of life descriptions and hypothetical product characteristics to gather their preferences for quality of life improvements and their willingness to pay for (unbranded) pain therapies at various levels of price. Pet owners were motivated to improve their cats’ quality of life, which translated into a willingness to pay for therapies; specifically, pet owners valued quality of life improvements in mobility, pain expression, and well-being. When presented with a product profile of the hypothetical new injection treatment and its cost, 50% of cat owners were willing to pay more for the new injection treatment, which is expected to have improved efficacy and safety when compared to a hypothetical standard treatment. Significantly more pet owners also preferred the new injection treatment to the standard treatment when the price was not presented (*p* < 0.01), with product efficacy and safety driving pet owners’ decision-making. The majority of pet owners did not agree that taking their cats to the veterinarian once a month for treatment would be burdensome. Cat owners in the UK are motivated to improve their cats’ quality of life, which translates into a willingness to pay for effective treatment of pain associated with osteoarthritis. Veterinarians should offer cat owners the pain treatment they feel is best suited for improving the cats’ quality of life and ensuring the relationship between cat and owner is preserved.

**Abstract:**

This research aimed to explore UK cat owners’ preferences for treatments for feline osteoarthritis (OA) by exploring preferences around quality of life (QoL) improvements, safety considerations, and costs associated with hypothetical innovative pain therapies. Aspects identified in an existing conceptual framework were extracted for inclusion in exploratory interviews with cat owners (*n* = 3) to identify key domains that contribute to the QoL of cats. QoL descriptions for cats with OA and hypothetical product attributes were developed and validated through interviews with veterinarians (*n* = 3). An online survey was subsequently shared with 255 pet owners in the UK. Pet owners were presented with QoL descriptions and hypothetical product attributes to gather their preferences for QoL improvements and their willingness to pay (WTP) for (unbranded) pain therapies at various price points. Pet owners were motivated to improve their cats’ QoL, which translated into WTP for therapies; specifically, pet owners valued QoL improvements in mobility, pain expression, and well-being. When presented with a product profile of the hypothetical novel monoclonal antibody (mAb) and cost, 50% of cat owners were willing to pay more for a mAb that is expected to have improved efficacy and safety when compared to a hypothetical standard of care (SoC). Significantly more pet owners preferred the mAb than the SoC when price was not presented (*p* < 0.01), with product efficacy and safety driving pet owners’ decision-making. The majority of pet owners did not agree that taking their cats to the veterinarian once a month for their treatment would be burdensome. Cat owners in the UK are motivated to improve their cats’ QoL, which translates into WTP for the efficacious treatment of pain associated with osteoarthritis. Veterinarians should offer cat owners the pain treatment they feel is best suited for improving the cat’s QoL and to ensure subsequent owner-pet bond is preserved.

## 1. Introduction

Osteoarthritis (OA) is a degenerative joint disease (DJD) that commonly affects cats and leads to gradual behavioral changes and decreased well-being as a result of chronic pain [1,2,3,4]. The prevalence of radiographic evidence of DJD has been reported to be 61–92% in cats [1,2,3]. Hardy et al. found radiographic evidence of DJD in 90% of cats; however, only 4% of the medical records of cats in the study had any mention of DJD or arthritis by the owner or the clinician [1].

The recognition of OA in cats by pet owners and veterinarians can be challenging [1,3]. OA is best identified by the cat owner in their home environment; however, cat owners often do not appreciate the fact that the changes they see in their cat are signs of OA [5]. Thus, OA in cats is largely underdiagnosed, meaning that many cats may not receive appropriate treatment for their condition, resulting in a reduction in their quality of life (QoL) [4,6]. Previous research has identified several key domains that are indicative of a cat’s QoL; these include domains such as energy and vitality, pain, mobility, and well-being [5,6,7]. When a screening checklist was used with cat owners, they were able to identify signs of OA pain, and this identification had a positive correlation with the diagnosis of OA [5].

The overall aim of pain management in the context of OA in cats is to achieve the best possible QoL outcomes for the cats and their owners [8]. Many interventions have been investigated for use in treating OA in cats, including pharmacological therapies such as nonsteroidal anti-inflammatory drugs (NSAIDs), tramadol, and gabapentin, along with other non-pharmacological treatment modalities such as nutraceuticals and environmental adaptations [8].

Pain assessment and subsequent management may result in multidimensional improvements in a variety of domains that can contribute to QoL [9]. Inadequate pain management may, therefore, result in a detrimental impact on a cat’s QoL, which may subsequently impact the owner-pet bond. Previous qualitative research has determined that the owner-pet bond is defined by specific behaviors related to the owners’ feelings about their pets, the amount of time they spend with the pet, and the activities they perform with their pets, alongside other factors [10].

It is currently not known which aspects of a cat’s QoL pet owners prioritize when considering the treatment of pain associated with OA, or which treatment features and benefits they favor when selecting treatment for their cat. Measuring willingness to pay (WTP) is an established methodology [11] that provides an approach for measuring pet owner preferences for QoL improvements and preferred treatment characteristics. In this case, QoL improvements could involve improvement of visual signs of distress or comfort and/or behaviors affecting the pet owner bond (such as reactivity or reluctance to interact). To ensure that cats suffering from pain associated with OA are able to achieve an acceptable QoL, clarification of pet owner preferences for improvements in QoL and treatment attributes is needed.

This research aimed to generate robust evidence for cat owners’ preferences around QoL improvements and potential innovative therapy profiles, as well as how these may affect WTP for pain associated with OA in cats. The outputs of this research aim to help facilitate both veterinarians and pet owners’ decision-making when considering pain management options.

## 2. Materials and Methods

### 2.1. Identification of QoL Domains of Importance to Cats and Their Owners

Insights from previous literature were combined with the results of prospective interviews conducted as part of this study in order to maximize the likelihood of reaching concept saturation during the development of scenarios and treatment profiles relating to feline OA.

Initial QoL domains were taken from a conceptual framework that was originally developed to assess health-related quality of life (HRQoL) in cats with OA [7]. The conceptual framework was derived from a systematic review of the published literature. Literature was identified from a series of medical and scientific databases using the search terms; “cat”; “feline”; “chronic pain”; pain; and “quality of life”. Publications were selected if they were full-text and peer-reviewed, based on primary data, and identified or measured behavioral signs of chronic musculoskeletal pain in cats [7]. The conceptual framework was developed according to published guidelines [7]. The domains extracted from the conceptual framework were therefore identified as key concepts for the HRQoL of cats with OA in a robust manner.

Following the extraction of previously identified domains, exploratory interviews were conducted with cat owners (*n* = 3) to explore the key domains identified in previous conceptual research and to identify any additional domains not covered that may contribute to feline QoL. The screening criteria for cat owners is presented in Table 1.

### 2.2. Cat Owner Interview Content

Cat owner interviews (in February–March 2020) included open-ended questions that aimed to spontaneously elicit their perception of the most important aspects of a cat’s QoL and the most important attributes of therapies when deciding about their cat’s care, in addition to probing on QoL domains that had been identified from the conceptual framework [7]. The most commonly reported mobility concepts mentioned by respondents when considering potential improvements in feline QoL were jumping, walking/gait, and running. The most commonly reported pain expression concepts mentioned by respondents when considering potential improvements in feline QoL were tolerance to touch and vocalization. The most commonly reported well-being concepts mentioned by respondents when considering potential improvements in feline QoL were appetite and sleeping habits. (Further details on the cat owner interview content can be found in the development phase: interviews with pet owners in the Appendix A).

### 2.3. Feedback Phase

Following the development interviews, draft QoL descriptions highlighting theoretical QoL improvements in cats with pain associated with OA were prepared for testing with veterinarians (*n* = 3). The content and distribution of QoL descriptions were reviewed by veterinarians to determine their appropriateness for inclusion in the survey. Veterinarians were probed on whether the QoL descriptions were an accurate representation of the QoL impairments experienced by cats with pain associated with OA. Veterinarians were screened according to the pre-defined criteria that are detailed in Table 2.

### 2.4. Veterinarian Interview Content

Qualitative interviews (in May 2020) were also conducted with veterinarians to test the domains for inclusion within QoL descriptions and important attributes of therapies for cat owner decision-making to take forward to the WTP fieldwork. Veterinarians’ responses to the attributes identified by pet owners suggested the following: inclusion of “stiffness” when referring to walking/gait, removal of vocalization, and addition of facial expression in pain expression; the addition of eating less due to poor mobility for appetite, and inclusion of willingness to explore outside for outdoor cats.

The veterinarians confirmed that the descriptions were an accurate reflection of the QoL of cats with pain associated with OA; however, the wording included within the descriptions was amended to better represent the language/terminology used by pet owners.

### 2.5. WTP Fieldwork

WTP fieldwork was conducted in August and September 2020. Following consolidation veterinarian feedback, a 45 min web-based survey was designed and piloted. For the initial pilot stage, 25 pet owners’ WTP was tested by asking them to consider their relative WTP for therapy for their cat when considering improvements in particular behavioral attributes to ensure the survey was easily understood by all respondents, as well as programmed appropriately.

Respondents had previously been registered with an online survey panel (Qualtrics, Provo, UT, USA) and had confirmed their interest in participating in the research. Respondents were contacted via email to participate in the survey, and a target sample size of 250 complete responses was sought, in line with previously published research [12,13,14,15,16,17,18,19,20,21,22,23,24]. The objective screening criteria applied to screen eligible participants are presented in Table 3.

Demographic data were collected, and quotas were applied for geographic regions to correspond with the distribution of the general UK population. Information relating to eligible respondents’ pets was also collected to ensure that key subgroups of interest (as identified during the qualitative interview stage) were present in the final sample (e.g., the presence of pet insurance and pet medical history).

All eligible respondents confirmed their willingness to participate and provided explicit informed consent. All respondents received financial compensation for their participation in the survey. The survey design is detailed in the Appendix A (see the Overview of WTP survey design in the Appendix A).

### 2.6. Testing WTP

An online survey link was shared with cat owners in August 2020 to investigate their preference when considering potential QoL improvements in cats with pain associated with OA. Cat owners’ WTP for QoL improvement was also explored in the survey.

Cat owners were presented with a series of QoL impairments and were asked to indicate which of the scenarios they would most like their cat to be in (Table 4). The statements were developed to reflect hypothetical improvements in specific QoL domains, including mobility, pain, appetite, grooming, sleep, and toileting behaviors. Cat owners were also asked to indicate how much they would be WTP for potential improvements in each wider QoL domain, comprising pain expression, well-being, and mobility.

The survey was live for three weeks to allow recruitment of cat owners to meet the minimum requirement for the study (*n* = 250, in line with previously published studies) [12,13,14,15,16,17,18,19,20,21,22,23,24].

Finally, cat owners were presented with hypothetical product profiles with varying product attributes and were asked which product profile they would prefer. Product profiles were highly realistic but were unbranded to avoid bias in participant responses (Table 5). As product profiles were hypothetical, cat owners were of the assumption that no label approval or pharmacovigilance data existed for both profiles. The hypothetical nature of product profiles was reiterated to the respondents. Once presented, respondents were asked to indicate how much they were WTP for each product profile, highlighting key drivers of cat owners’ WTP.

### 2.7. Data Analysis

Survey results were collated, quality checked, and analyzed. Demographic data and WTP data were analyzed descriptively. 

The statistical significance of differences was assessed using a threshold of *p* < 0.05.

Anonymized results data are available from the corresponding author upon reasonable request.

## 3. Results

The outputs of the qualitative development and feedback stages of this research are detailed in the Appendix A.

### 3.1. Sample Characteristics

A total of 255 cat owners participated in the WTP survey. The recruited samples were geographically representative of the UK population and covered a wide range of age groups and sociodemographic backgrounds (Table 6). 

Demographic information regarding the respondents’ current pets was also collected (see Table 7). Overall, 58% of the respondents had one cat within their household. A total of 8% of the overall sample had a cat living with pain associated with OA, with the majority (65%) of those receiving medication for their condition. Of the 255 pet owners who did not currently have a cat living with pain associated with OA, a further 25 (11%) had experience of this in the past.

### 3.2. WTP for QoL Improvements in Pain Associated with OA

Overall, 86% (*n* = 219/255) of the respondents answered that they would be willing or very willing to initiate any treatment for their cat when considering pain associated with OA.

When presented with wider QoL domains, consisting of pain expression, mobility, and well-being, approximately half of all pet owners were willing to pay for improvements in each of the wider domains listed (Table 8).

When presented with a series of QoL trade-offs among more specific domains, cat owners appeared to value QoL across all specific domains tested; relative preferences for pain expression, mobility, and well-being appeared equal across specific domains (Figure 1).

Pet owners’ WTP for QoL improvement was also demonstrated through a rating exercise. The majority of pet owners (61%; *n* = 155/255) answered that they would be willing to pay more to ensure their cat has an improved QoL. This statement was rated highly (mean rating of 5.58, on a scale from 0.00 [indicating “strongly disagree”] to 7.00 [indicating “strongly agree”]), indicating a high level of willingness by cat owners.

### 3.3. Treatment Profile Preferences

When presented with monoclonal antibody (mAb) vs. SoC profiles, significantly more pet owners preferred the mAb profile (72%; *n* = 184/255) vs. the SoC (28%; *n* = 71/255) when the price was not presented (*p* < 0.01). When presented with price, 50% (*n* = 127/255) of cat owners answered that they would be willing to pay more than the SoC (priced at £30 [$40.50/€33.60] per month) for the mAb (priced at £79 [$106.65/€88.48] per month), despite the increase in price (Figure 2).

When asked unaided how much more they would be willing to pay for the mAb, over half of all pet owners answered that they would be willing to pay an average of £27 [$36.45/€30.24] more than the SoC price per month for the mAb profile. Price estimates ranged from £5 [$6.75/€5.60] more per month to £150 [$202.50/€168] more per month.

### 3.4. Administration Profile Preferences

In a series of ranking statements, 79% (*n* = 201/255) of cat owners strongly agreed or agreed that they would rather administer the treatment themselves to the cats than visit the veterinarian for administration (mean rating of 6.12/7.00). However, when presented in the context of a product profile, the safety and efficacy of a therapy appeared to supersede the method of administration. Despite cat owners rating home administration highly, respondents stated that they would prioritize the safety and efficacy of therapy when considering factors of importance for cat owner decision-making, in the hypothetical scenario that was presented (Figure 3).

Additionally, when asked to rate whether cat owners would find taking their cat to the veterinary clinic once a month stressful/burdensome for their treatment, 85% (*n* = 217/255) of cat owners did not strongly agree that they would find the trip to the veterinarians once a month stressful/burdensome (mean rating of 3.92/7.00). Cat owners also declared that they would be comfortable with both a veterinarian and veterinary nurse administering their treatment to their cat; significantly more cat owners (86%; *n* = 219/255) said that they would prefer a veterinary nurse to administer a treatment to their cat at a lower cost per month (presented at £67.50 [$91.30/€75.60] per month) when compared to veterinarian administration (14%; *n* = 38/255, *p* < 0.01; presented as £79 [$106.65/€88.48] per month).

## 4. Discussion

Although OA is often underdiagnosed and subsequently undertreated in cats [4], this study estimated that over 80% of cat owners in the UK would be willing or very willing to initiate any treatment for their cats when considering the management of pain associated with OA. Additionally, the majority of pet owners did not agree that taking their cat to the veterinarian once a month for their treatment would be stressful or burdensome. Pet owners valued QoL improvements across key behavioral domains such as mobility, pain expression, and well-being.

Moreover, pet owners said that they were willing to pay an average of £27 [$36.45/€30.24] more than the SoC per month for innovative treatment, with an expected improved efficacy and safety profile, if available. This suggests that pet owners value their cats’ QoL and would be willing to initiate and pay more for innovative therapy options to ensure their cat’s QoL improved. Pet owners do not prioritize the administration methods of therapies over their safety and efficacy. Pet owners appear comfortable with having an injection administered by a veterinarian or nurse once a month or administering treatments themselves to their cats.

These findings build upon a previous evidence-based conceptual framework of the factors influencing HRQoL in cats with OA [7]. This framework presented three wide HRQoL domains (namely pain expression, mobility, and physical and mental well-being), each of which was subsequently tested in the current research in terms of pet owners’ WTP. Our findings further suggest that varied preferences exist for efficacy aspects and product profiles in the treatment of pain related to feline OA in the UK. Veterinarians in the UK could therefore elect to offer cat owners the pain treatment they feel is best suited for improving cat QoL, to ensure QoL of the cat and owner-pet bond is preserved.

To our knowledge, this research is one of the few WTP studies conducted on companion animal health [12]. The WTP methodology is a widely used process that is utilized to understand how stakeholders value a proposed product or concept [11]. The application of this trusted methodology within animal health has provided insight into key drivers in pet owner decision-making, when considering the preservation of their cat’s QoL and the subsequent owner-pet bond [10,11].

At present, there are limited NSAID therapies licensed for the treatment of chronic pain associated with OA in cats, and licensing differs across the globe. In an online questionnaire with cat owners (*n* = 46), only 54% of drugs used to treat felines for a variety of conditions were registered for use in cats by the European Medicines Agency (EMA); while this research was not solely focused on the management of chronic pain associated with OA, these findings emphasize the frequency of off-label drug use in the feline veterinary setting [25]. Additionally, cat owners emphasized the need for more easily administered therapies for the treatment of feline conditions; very few (*n* = 3/43) cat owners said that their cat consumed the medication willingly without requiring any modifications to the dosage form or administration aid [25]. Consequently, the individual nature of the cat should be taken into account in prescribing decisions [25]. The need for more easily administered therapies and difficulties related to administration highlights a desire from pet owners for additional licensed and easily administered therapy options, with established evidence supporting the efficacy and safety of the products’ use [25].

This research suggests that pet owners in the UK are motivated to improve cat QoL by initiating innovative therapies for pain associated with osteoarthritis. It is important to consider that all product profiles were hypothetical in this research; thus, there was an assumption by respondents that the weight of evidence and approval status for both drugs were the same, as no evidence was presented to suggest otherwise. Therefore, pet owner preferences for real (not hypothetical) treatment options should be interpreted in the context of the available data on efficacy and safety at the time of reading. The use of hypothetical product profiles is a common approach in human and animal health preference research [17,26,27,28,29].

Additionally, while these findings represent a currently unmet need in the UK for further licensed and easily administered therapy options, the generalizability of this research may be limited when considering its validity in other geographic regions. Future research may wish to repeat the methodology employed in this study in other geographic areas, such as the US, where labeling details for the SoC vary compared to those in the EU.

Additionally, the limited treatment options for pain associated with OA will likely have constrained pet owners’ experiences when considering the successful management of pain associated with OA. Thus, the lack of adequate and appropriate treatment options may have limited pet owners’ ability to predict how much they would be WTP for an innovative therapy.

Finally, this research was conducted in 2020 during the COVID-19 pandemic. Given that this research aimed to explore pet owners’ WTP for innovative therapies for their cats, our findings may have been impacted by the financial uncertainty associated with the pandemic. The Office for National Statistics demonstrated that the COVID-19 pandemic subsequently led to an abrupt reduction in household income, as real disposable income decreased by 2.3% during the first UK lockdown. The exacerbation of financial concerns may have impacted the WTP indicated by pet owners in this research. Future researchers may, therefore, wish to re-conduct a similar study to examine whether pet owners are WTP for innovative therapies at times of financial stability.

Although demographic factors such as income and insurance status may interact with and influence respondents’ preferences, no adjusted analysis was conducted as part of this research. However, this study was primarily focused on estimating the preferences of UK cat owners as a whole, which was achieved through the basic unadjusted analysis presented here. In addition, the reported household income of the sample appeared broadly representative of the UK general public (with 68% of respondents reporting a household income below £40k), and pet insurance was held by approximately half of the respondents. Future research may wish to specifically address potential interactions between factors such as income and insurance status in terms of their effect on pet owners’ WTP for treatment.

While the above concerns are valid, the sample size of this research (*n* = 255 quantitative survey responses) is generally consistent with those of other discrete-choice studies on animal and human health, which typically analyze 100 to 500 responses [12,13,14,15,16,17,18,19,20,21,22,23,24].

## 5. Conclusions

Pet owners in the UK value treatment profiles that are efficacious in improving QoL and deemed appropriate for long-term use. Veterinarians should, therefore, discuss all licensed treatment options with pet owners and recommend the treatment they believe is best suited, balancing aspects such as effectiveness, tolerability, convenience, and cost. This will help to ensure the treatment outcomes and QoL improvements cat owners desire are achieved and the owner-pet bond is preserved.

## Figures and Tables

**Figure 1 animals-14-02308-f001:**
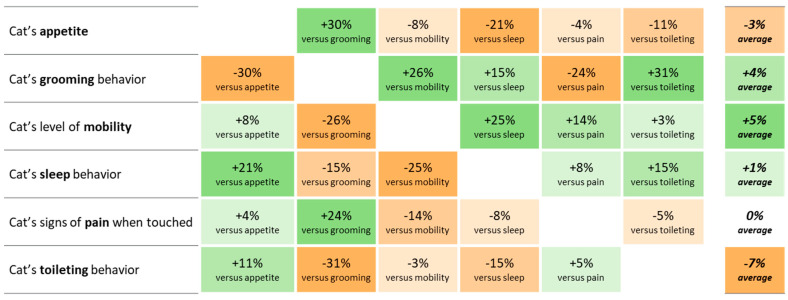
Pet owner survey respondents’ rate of prioritization of attributes from QoL trade-offs (*n* = 255).

**Figure 2 animals-14-02308-f002:**
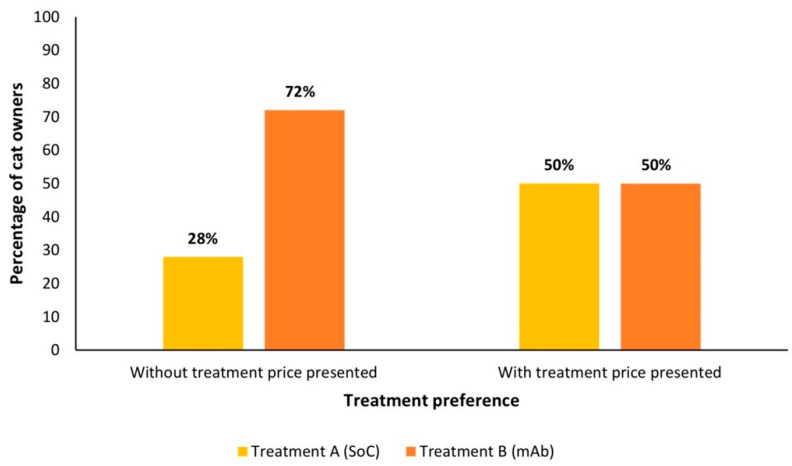
Pet owner survey respondents’ preferences for SoC vs. mAb, with and without treatment prices (*n* = 255). mAb, innovative monoclonal antibody treatment. SoC, standard of care.

**Figure 3 animals-14-02308-f003:**
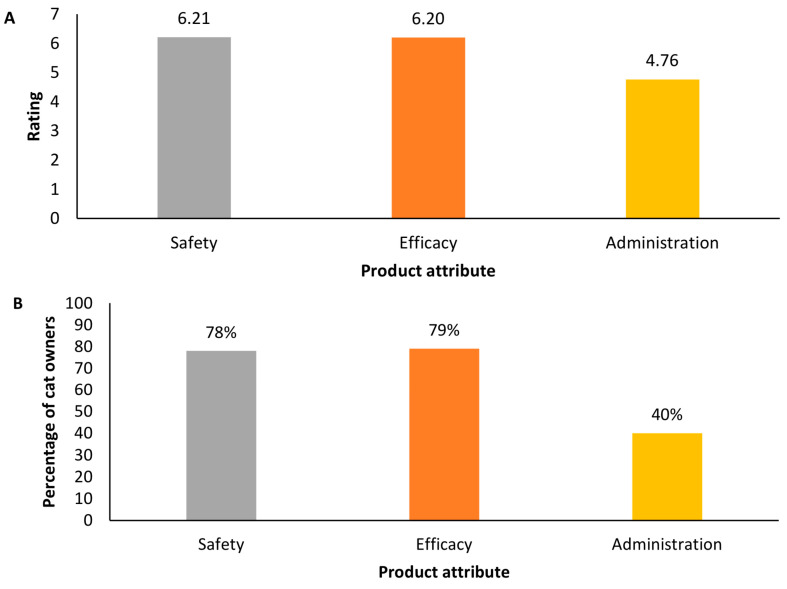
Pet owner survey respondents’ (*n* = 255) mean importance rating for product attributes (with one being least important and seven being most important; (**A**)) and the percentage of pet owners rating attributes as important or most important (**B**).

**Table 1 animals-14-02308-t001:** Cat owner recruitment criteria for qualitative development interviews prior to survey development.

Cat Owners Were Required to Be:
Aged ≥18, resident in the United Kingdom (UK)
Willing to give informed consent to participate
An owner of a cat with chronic pain associated with OA diagnosed by a veterinarian at least six months previously
Primarily or jointly responsible for decisions regarding healthcare products for their cat’s treatment
In contact with a veterinarian at least once in the past year in relation to the treatment of chronic pain associated with OA for their cat
Owned no more than five cats and dogs in total (combined)
Not employed by an advertising agency, market research firm, manufacturer or retailer of animal health products, professional breeder of cats or dogs, or a veterinary clinic

OA, Osteoarthritis.

**Table 2 animals-14-02308-t002:** Veterinarian recruitment criteria for qualitative validation interviews prior to survey development.

Veterinarians Were Required To:
Have been in the UK for >2 years
Personally, care for at least 30 cats in an average week and 120 in an average month
Have seen >0% of cats that had been diagnosed with (or were suspected to be experiencing) chronic pain due OA
Responsible for prescribing therapies for cats’ chronic pain due to OA (individually or as part of a group)
Not employed by an advertising agency, market research firm, manufacturer or retailer of animal health products, regulatory body or government institute, professional breeder of cats or dogs, or a university or academic veterinary practice

OA, Osteoarthritis.

**Table 3 animals-14-02308-t003:** Recruitment criteria for survey respondents.

Eligible Participants Were Required To:
Be adult (≥18 years) residents in the UK
Be owners of at least one pet cat (no more than five dogs and/or cats in total in household)
Be solely or jointly responsible for decision-making regarding veterinary products for their cats
Demonstrate adequate literacy in terms of veterinary information. ^a^

^a^ Adequate literacy was defined as a score of 4 or 5 on each of three questions on self-reported reliance on help when reading veterinary materials, confidence filling out veterinary forms, and frequency of problems reading veterinary materials. A score of 5 represented the lowest reliance on help, highest confidence, and lowest frequency of problems.

**Table 4 animals-14-02308-t004:** Example presentation of trade-offs between specific QoL domains to survey respondents.

A		B
Your cat… …has good mobility but …reacts negatively when touched	vs.	Your cat… …has reduced mobility but …does not react negatively when touched
Your cat… …does not react negatively when touched but …shows deterioration in sleep	vs.	Your cat… …reacts negatively to touch but …shows no changes in sleep
Your cat… …has good mobility but …shows deterioration in sleep	vs.	Your cat… …has reduced mobility but …shows no changes in sleep
Your cat… …has good mobility but …shows deterioration in grooming habits	vs.	Your cat… …has reduced mobility but …shows no changes in grooming habits
Your cat… …has good mobility but …shows deterioration in toileting behavior	vs.	Your cat… …has reduced mobility but …shows no changes in toileting

**Table 5 animals-14-02308-t005:** Example presentation of hypothetical product profiles to survey respondents.

**Please imagine the following situation**: Your veterinarian advises that your cat is suffering from pain associated with arthritis and recommends that your cat should receive treatment. Your veterinarian offers you a choice of two treatment options, which have different characteristics.	
**Please select the treatment that you would prefer to receive:**	
Product A	Product B (innovative therapy)
An oral solution given each day that you can give to your cat via a dosing syringe onto your cat’s food or by emptying the syringe directly into your cat’s mouth. This treatment can provide short-term pain relief and improve your cat’s mobility.	An injection that is given once a month to your cat at your vet clinic.
This treatment is not recommended for long-term use for pain relief and has a label warning against use in cats with kidney disease.	This treatment can provide long-term pain relief and improve your cat’s mobility This treatment can be used long-term and is well-tolerated in cats with kidney disease.
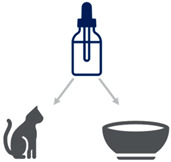	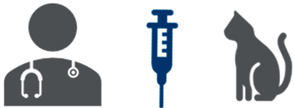
☐ Treatment A	☐ Treatment B
#18b: If treatment A costs £30 per month, what would you be willing to pay for treatment B?	
☐ I would not be willing to pay any more than £30 per month.	☐ I would be willing to pay £30 + ___________ per month for treatment B

**Table 6 animals-14-02308-t006:** Demographic characteristics of survey respondents.

Pet Owner Characteristic	*n*, (%) (*n* = 255)
Sex, *n* (%)	
Male	64 (25)
Female	191 (75)
Age range groups, *n* (%)	
18–30	46 (18)
31–40	53 (21)
41–50	68 (27)
51–60	45 (18)
>60	43 (17)
Household income, *n* (%)	
<20 k	82 (32)
20–40 k	91 (36)
40–60 k	46 (18)
60–80 k	21 (8)
>80 k	15 (6)
Geographical location, *n* (%)	
London	34 (13)
North East/North West/Yorkshire and Humberside	61 (24)
Scotland/Wales/Northern Ireland	41 (16)
South West/South East/East	82 (32)
West Midlands/East Midlands	38 (15)

**Table 7 animals-14-02308-t007:** Characteristics of survey respondents’ household pets.

Pet Characteristics	*n*, (%) (*n* = 255)	
Number of household pets	Cats	Dogs
1	148 (58)	58 (23)
2	82 (32)	11 (4)
3	12 (5)	7 (3)
4	9 (4)	0 (0)
5	4 (2)	0 (0)
Insurance		
Pet owners with cat insurance	114 (45)	
Pet owners without cat insurance	141 (55)	
OA diagnoses		
Pet owners with cats currently diagnosed with OA	20 of 255 total (8)	
Mean time since diagnosis	46 months	
Pet owners with a cat that had been diagnosed with OA previously	25 of 235 without current OA (11)	
OA treatments		
Meloxicam	10	
Tramadol	7	
Gabapentin	4	
Ketoprofen	4	
Amantadine	2	
Robenacoxib	1	
Buprenorphine	1	

**Table 8 animals-14-02308-t008:** Pet owners survey respondents’ WTP to gain improvements in additional wider QoL domains (*n* = 255).

Wider Domain of QoL	Number of Respondents Willing to Pay More Than for the SoC, in Order to Additionally Resolve This Wider Domain of QoL	Average Price More Than SoC Respondents Were Willing to Pay ^b^
Pain expression	119 (47%) ^a^	£30.00 ($40.50/€33.60)
Well-being	116 (45%) ^a^	£29.00 ($39.15/€32.48)
Mobility	113 (44%) ^a^	£31.00 ($41.85/€34.72)

^a^ No statistical differences were identified via chi-square in the number of respondents WTP to additionally improve pain (*p* = 0.29), well-being (*p* = 0.15), and mobility (*p* = 0.07) when compared to those who were not WTP to pay anything in addition to the SoC. ^b^ Rounded to nearest whole number; price presented is how much more pet owners were WTP above SoC (i.e., the SoC cost of £30 [$40.50/€33.60], plus the average additional price respondents were WTP to resolve this additional wider domain of QoL). QoL, Quality of life; SoC, Standard of care.

## Data Availability

Data for this study will be made available upon reasonable request.

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
