# Peer review of "Pet Owners’ Preferences for Quality of Life Improvements and Costs Related to Innovative Therapies in Feline Pain Associated with Osteoarthritis—A Quantitative Survey"

_animals, 2024, doi:10.3390/ani14162308_

Round 1

Reviewer 1 Report

Comments and Suggestions for Authors

This research is very interesting and important. It uses an method that is not often used in animal-research but provides interesting information about owners’ ideas about treatment and QoL of their companion animals.

As I had no access the figures and supplemental materials I could no review these.

I have some minor remarks and some questions

Introduction

Line 70. Consider switching the last two sentences of this paragraph.

Line 87. Some examples on the owner-pet bond could be provided such as pain-related aggression.

Line 99. Consider omitting ‘to enable the owner-pet bond to be maintained’. This seems not relevant as the sentences starts with To ensure cats suffering from pain achieve etc.

Methods

I wonder why exclude owners with more than 5 pets

The number of interviews seem rather low, why n=3? And not n = 5?

Line 188. I would like to see information that the sample is indeed representative of the UK cat population.

Line 206: I do not understand how the trade-offs mentioned in Table 4 relate to the scenario’s displayed in the survey. Could you elaborate more on this? Maybe write down one of the scenaripo’s as you did with the products inTable 5.  

Product A does resemble Novacam administration although the authors claim the products are hypothetical.

Results

Please provide more details how you came to the three groups presented in Table 8 (Pain Expression, Well-being and Mobility).

Line 286. I do not understand the last sentence: ‘This statement was rated highly when considering cat owners’ level of agreement. Which level of agreement?

Did you also ask if owners would pay more if the administration could be done at their house?

If treatments A and B are hypothetical why is it relevant to mention the monoclonal antibody vs the SoC? Why not speak of treatment A vs treatment B. The study is not about the difference between the current treatment and willingness to try a novel treatment or am I mistaken?  

8% of cats in sample had OA related pain, 68% received medication. I wonder if specific sample characteristics are associated with yes/no medication (for example income)?

Conclusions

I fully agree with the conclusions.

Conflict of interest

Although potential conflict of interests are declared. Are the authors involved in development of novel pain treatments?

The last sentence in the conflict of interest section seems odd.

Reviewer 2 Report

Comments and Suggestions for Authors

Lines 39-52 are quite duplicative from the simple summary. Please make this abstract more distinctive with more detail.

Line 80 and others: pet-owner bond: I also believe that this relationship and better understanding of owner preferences can be invaluable for veterinarians in their efforts to do this. Only line 104 mentions the role of veterinarians in helping owners help their cats improve their QoL.

Table 1: how were cat owners identified to be screened? Please add briefly.

Lines 122: please include the prompting questions either here or in the supplement.

Lines 125-6: I believe that reference 33 here is the published one in 2021 with an edited title? The reference should also precede the parenthetical section for clarity (the additional info in the supplement is in this manuscript not the published paper).

Table 2: again, how were these veterinarians identified to be screened?

Line 142: please share a bit more about what questions were asked or how information was elicited either here or in the supplement especially how the reaction to QoL descriptions was handled.

Line 156: it isn’t until the ethical paragraph that we find out an outside firm did the recruitment. Please add details here including how the survey was described to participants as they can influence their responses. Is the survey itself available? If so, it would be very helpful to include in the supplement—I see the process and topics but not the actual questions.

Line 173: how much were they paid?

Supplement for WTP section: I see toileting in Table 2/4 but not listed in Table 1/3. Is that correct that owners didn’t identify toileting as a problem associated with OA. Please clarify in text.

Line 182-3: the prior work identified 7 domains. I’m assuming that the authors are using the 3 from the conceptual framework rather than the 7 from the domains list in that publication? That needs to be explored a bit more clearly in the intro and then listed specifically in the methods so that it is clear how this all fits together. Line 205: it isn’t clear here which are the “main domains” from the conceptual framework, and which are the 7 “domains” without reading the reference pretty closely. Please clarify in the text and list all 7 and the main or key domains and what is included. In table 8 and accompanying text, only 3 domains are listed but the examples of trade offs seem to come from all 7 domains (the figure 3 in the reference)? Please clarify in text.

Line 208-9: what subgroups were these? Please list in the manuscript.

Table 7: Is time to diagnosis normally distributed? If not, please use median and min/max. For “pet owners diagnoses with OA previously”: are these an additional 25 owners to the current cats? It appears that way based on the denominator but please clarify in text.

Table 8: I don’t understand what proportions are being compared for each of the rows for the number of respondents. I only see data for # willing to pay more (119)  vs # willing to pay less (136). Subscript b has an error in the text. But also, is the £30 the SoC for pain or is that how much over the original SoC of £30 they were willing to pay—so for pain a total of £60? Please clarify as I believe that is the case, but it is confusing especially with the extra stuff in [] (then in dollars and euros which is missing the explanation—also in the methods that these were calculated based on something at a point in time I’m assuming.). Anova: What is the single factor? There are 3 domains. And the WTP in £ for the dependent variable? That isn’t clear in the methods or here. Were the assumptions for ANOVA met? Please add to methods and here.

Line 247-8: please be clear about the differences between “main domains” and attributes and other domains…it is confusing.

Line 253-5: I don’t see this listed in the methods. The response options should be included there. Here indicate also the min/max of ratings. Also line 276 and following.

Figure 2: there are two symbols which are not defined. Please add to text.

Note that in several spots including line 272 and 302 there is the numeral 2.

Table titles and figure legends should be expanded to be standalone and include information on the study and sample. Please also clearly label the main and supplementary tables and reference each in the text where they are most appropriate.

Discussion: I also think that these results are quite interesting because the alternative is a monoclonal antibody…there is (at least in some countries) some concerns over anything that might be “genetically modified” or related to a vaccine. Clearly, it depends on how the mAb was presented (which should be explained in a bit more detail) but this is encouraging that the newer technology wasn’t seen to be an issue with adoption.  I would like to see something in the discussion to this effect.

Line 304: that they don’t prioritize administration over safety and efficacy seems to be a socially acceptable set of responses. Please add some additional modifying language when discussing the findings since this is a hypothetical situation and owners are “reporting” what they believe, and we know that what people believe and what they do are not the same.

Paragraph starting on line 341: Were the other demographic variables similar to the population as a whole? That should be in the results for clarification.

Conclusions: I’d love to see something a bit more here about what to consider for “best suited” given the interest internationally in contextual medicine and spectrum of care and providing what is appropriate for that cat and person/family. Just a little bit more.

Reviewer 3 Report

Comments and Suggestions for Authors

Thank you for the opportunity to review your manuscript. This was a survey study assessing the preferences of pet owners regarding the potential for improving quality of life and the costs associated with innovative therapies to treat pain associated with osteoarthritis in cats. The study provided hypothetical treatment options to compare owner preferences for standard of care medications versus treatment with monoclonal antibody for pain management. Key findings identified included that cat owners were more willing to pay for monoclonal antibody treatment if it was expected to have improved efficacy and safety versus standard of care treatment options, and that cat owners did not find taking the cat to the veterinarian once a month for treatment would be a significant burden. The authors note that the study was conducted during the pandemic and therefore the cost of treatment did alter pet owner’s preferences to choose one hypothetical treatment over the other, although more owners would choose the innovative treatment option still given the potentially improved safety and efficacy. The manuscript was clearly written and easy to understand. 

I have just one comment for consideration to improve the manuscript. The simple summary is quite lengthy and contains much of the same details as the abstract. According to the author guidelines provided by Animals, the simple summary should be simple and concise, written for a lay audience and should not contain abbreviations.
